# Maternal weight change from prepregnancy to 18 months postpartum and subsequent risk of hypertension and cardiovascular disease in Danish women: A cohort study

Helene Kirkegaard[1,2]*, Mette Bliddal[2], Henrik Støvring[3], Kathleen M. Rasmussen[4], Erica P. Gunderson[5], Lars Køber[6], Thorkild I. A. Sørensen[7], Ellen A. Nøhr[1,8]

1 Research Unit of Obstetrics and Gynecology, Department of Clinical Research, University of Southern Denmark, Odense, Denmark, 2 Open Patient Data Explorative Network (OPEN), Department of Clinical Research, University of Southern Denmark and Odense University Hospital, Odense, Denmark, 3 Department of Public Health, Biostatistics, Aarhus University, Aarhus, Denmark, 4 Division of Nutritional Sciences, Cornell University, Ithaca, New York, United States of America, 5 Division of Research, Cardiovascular and Metabolic Conditions Section, Kaiser Permanente Northern California, Oakland, California, United States of America, 6 Department of Cardiology, Rigshospitalet, University Hospital, Copenhagen, Denmark, 7 Novo Nordisk Foundation Center for Basic Metabolic Research and Department of Public Health, Faculty of Health and Medical Sciences, University of Copenhagen, Copenhagen, Denmark, 8 Department of Gynecology and Obstetrics, Odense University Hospital, Odense, Denmark

* hkirkegaard@health.sdu.dk

**Data Availability Statement:** The data that the findings of this study are based on came from the

## Abstract

### Background

One-fourth of women experience substantially higher weight years after childbirth. We examined weight change from prepregnancy to 18 months postpartum according to subsequent maternal risk of hypertension and cardiovascular disease (CVD).

### Methods and findings

We conducted a cohort study of 47,966 women with a live-born singleton within the Danish National Birth Cohort (DNBC; 1997–2002). Interviews during pregnancy and 6 and 18 months postpartum provided information on height, gestational weight gain (GWG), postpartum weights, and maternal characteristics. Information on pregnancy complications, incident hypertension, and CVD was obtained from the National Patient Register. Using Cox regression, we estimated adjusted hazard ratios (HRs; 95% confidence interval [CI]) for hypertension and CVD through 16 years of follow-up. During this period, 2,011 women were diagnosed at the hospital with hypertension and 1,321 with CVD. The women were on average 32.3 years old (range 18.0–49.2) at start of follow-up, 73% had a prepregnancy BMI <25, and 27% a prepregnancy BMI ≥25. Compared with a stable weight (±1 BMI unit), weight gains from prepregnancy to 18 months postpartum of >1–2 and >2 BMI units were associated with 25% (10%–42%), P = 0.001 and 31% (14%–52%), P < 0.001 higher risks of hypertension, respectively. These risks were similar whether weight gain presented postpartum weight retention or a new gain from 6 months to 18 months postpartum and whether GWG was below, within, or above the recommendations. For CVD, findings differed according to

Danish National Birth Cohort, and restrictions apply to these data, which were used under license for the current study and are not publicly available. The Danish National Birth Cohort welcomes requests for data which must include a short protocol with a specific research question and a plan for analysis. More information can be found at www.dnbc.dk.

**Funding:** The Danish National Birth Cohort was established with a significant grant from the Danish National Research Foundation. Additional support was obtained from the Danish Regional Committees, the Pharmacy Foundation, the Egmont Foundation, the March of Dimes Birth Defects Foundation, the Health Foundation and other minor grants. Follow-up of mothers and children have been supported by the Danish Medical Research Council (SSVF 0646, 271-08-0839/06-066023, 0602-01042B, 0602-02738B), the Lundbeck Foundation (195/04, R100-A9193), The Innovation Fund Denmark 0603-00294B (09-067124), the Nordea Foundation (02-2013-2014), Aarhus Ideas (AU R9-A959-13-S804), University of Copenhagen Strategic Grant (IFSV 2012), and the Danish Council for Independent Research (DFF – 4183-00594 and DFF - 4183-00152). The corresponding author, Helene Kirkegaard, received a grant from the Danish Heart Foundation (14-R97-A5163). The foundations were not involved in the conduct of the study, analysis and interpretation of the results or preparation, review, or approval of the manuscript.

**Competing interests:** The authors have declared that no competing interests exist.

**Abbreviations:** CI, confidence interval; CVD, cardiovascular disease; DNBC, Danish National Birth Cohort; GDM, gestational diabetes mellitus; GWG, gestational weight gain; HR, hazard ratio; ICD, International Classification of Diseases; IOM, Institute of Medicine; LDL, low-density lipoprotein; STROBE, Strengthening the Reporting of Observational Studies in Epidemiology.

prepregnancy BMI. In women with normal-/underweight, weight gain >2 BMI units and weight loss >1 BMI unit were associated with 48% (17%–87%), $P = 0.001$ and 28% (6%–55%), $P = 0.01$ higher risks of CVD, respectively. Further, weight loss >1 BMI unit combined with a GWG below recommended was associated with a 70% (24%–135%), $P = 0.001$ higher risk of CVD. No such increased risks were observed among women with overweight/obesity (interaction by prepregnancy BMI, $P = 0.01$, 0.03, and 0.03, respectively). The limitations of this observational study include potential confounding by prepregnancy metabolic health and self-reported maternal weights, which may lead to some misclassification.

## Conclusions

Postpartum weight retention/new gain in all mothers and postpartum weight loss in mothers with normal-/underweight may be associated with later adverse cardiovascular health.

## Author summary

### Why was this study done?

- Many women experience persistent weight gain from childbearing. This pregnancy-related weight change may be associated with worse long-term cardiovascular health.

### What did the researchers do and find?

- We used data from 47,966 mothers who participated in the Danish National Birth Cohort (DNBC).

- Self-reported weights were used to define their weight change patterns from prepregnancy to 6 and 18 months postpartum. We examined how these patterns were related to their risk of hypertension and cardiovascular disease (CVD) the following 16 years.

- We found that weight gain from before pregnancy to 18 months postpartum was positively associated with the risk of hypertension regardless of whether the women retained weight from pregnancy or gained weight from 6 to 18 months postpartum.

- In women with normal-/underweight, risk of CVD increased with a weight gain from before pregnancy to 18 months postpartum, but also with a weight loss in this period, especially if they had gained below recommended during pregnancy. No such increased risks of CVD were observed in women with overweight/obesity.

### What do these findings mean?

- Our findings suggest that health professionals should also focus on the mother's weight change patterns after given birth to improve their cardiovascular health. While women with overweight should avoid weight gain, both weight gain and loss should be of concern among women with normal-/underweight.

## Introduction

Cardiovascular disease (CVD) is the most common cause of death in European women [1], with hypertension as an important risk factor [2]. Although cardiovascular mortality in general has decreased steeply over the past decades, this has not been observed in young women [3]. Childbearing is common in young adulthood, and for many women, childbearing is related to a persistent weight gain; almost one-fourth experience a substantial higher weight 1 to 2 years after delivery than before pregnancy (>4.5 kg) [4–6]. This weight gain increases risk of obesity later in life [7,8], an important risk factor for hypertension and CVD [9–11]. Moreover, the additional weight postpartum may lead to a proportional increase in abdominal adiposity [12,13] which is highly correlated with adverse cardiovascular health [14,15]. Thus, maternal weight changes throughout pregnancy and after birth may be of great importance for women's long-term cardiovascular health.

Although women may have the same overall weight change from before pregnancy to postpartum, the patterns differ. A higher weight postpartum may result from a retention of gestational weight gain (GWG) or new weight gain in early motherhood [5], and a lower weight postpartum may result from low GWG or weight loss in early motherhood [16]. These patterns may have different causes and underlying metabolic mechanisms and therefore have different potential influences on mothers' subsequent cardiovascular health. Moreover, more than 30% of Danish women [17] and almost 50% of the United States women [18] live with overweight or obesity when they become pregnant, and for these women, a substantially higher weight postpartum than before pregnancy may exacerbate an already elevated risk of hypertension and CVD [19,20].

We aimed to examine how weight change from prepregnancy to 18 months postpartum was associated with subsequent maternal risk of incident hypertension and CVD, while considering prepregnancy BMI as well as the respective contributions of GWG and postpartum weight change patterns to the overall weight change.

## Methods

### Danish National Birth Cohort

The Danish National Birth Cohort (DNBC) enrolled 91,381 pregnant women between 1996 and 2002. Detailed description of the cohort has been reported previously [21]. Briefly, women were invited to participate at their first antenatal visit at their general practitioner. Four telephone interviews were carried out, 2 during pregnancy (approximately at week 16 and 30) and 2 postpartum (approximately 6 and 18 months after birth). A food frequency questionnaire was filled out approximately at pregnancy week 26, covering the previous month's dietary intake. Furthermore, approximately 14 years after delivery, a maternal follow-up was conducted with a participation rate of 53% [22]. All questionnaires are available at https://www.dnbc.dk/data-available.

### Study population

We included 86,209 women with their first pregnancy ending in a live birth within the DNBC as the index pregnancy. Women were excluded due to occurrence of the following before start of follow-up (18 months postpartum): death (*n* = 27); emigration (*n* = 571); and any diagnosis of either hypertension, ischemic heart disease, stroke, or other CVDs (*n* = 849) (Fig 1). Furthermore, 3,268 women who had had a subsequent birth and 9,189 women who were pregnant (>1-month duration) within 18 months postpartum were excluded as this may have affected their weight. Also, 5,743 women who were missing prepregnancy BMI values were excluded as

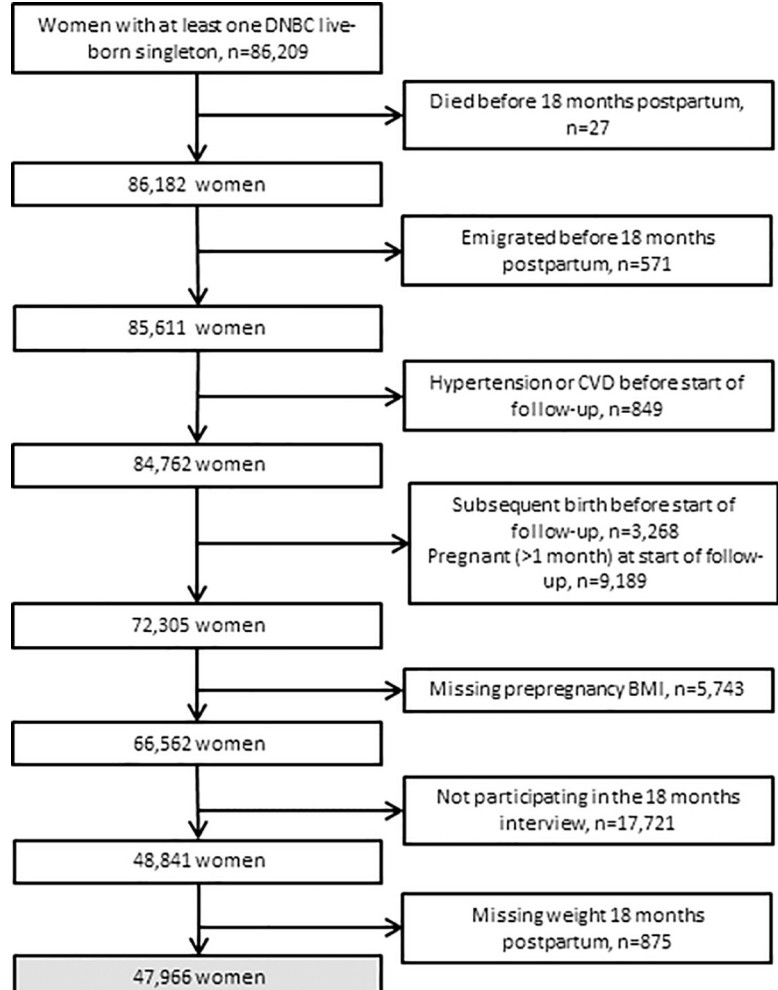

**Fig 1. Flowchart of the study population.** CVD, cardiovascular disease; DNBC, Danish National Birth Cohort.

were women who did not participate in the 18 months postpartum interview (17,721 women) or did not provide weight at that time (875 women). Thus, our final study population included 47,966 mothers.

All participants provided written informed consent. The DNBC was approved by the Scientific Ethic Committee in Denmark and the Danish Data Protection Agency, which also approved the present study. This study is reported according to the Strengthening the Reporting of Observational Studies in Epidemiology (STROBE) guideline (S1 STROBE Checklist).

## Weight

All weight information was self-reported. From the first pregnancy interview, we had information on prepregnancy weight and height, and at the interview 18 months postpartum, the women provided their current weight. Our main exposure was change in BMI (weight (kg)/ height (m)$^2$) from prepregnancy to 18 months postpartum, which allowed us to take into account both height and overall body size. For a woman of 1.68 m, a 1-unit increase in BMI corresponded to a weight gain of 2.82 kg. We divided the women into 4 groups: $<-1$, $-1$ to 1, $>1$ to 2, and $>2$ unit changes in BMI.

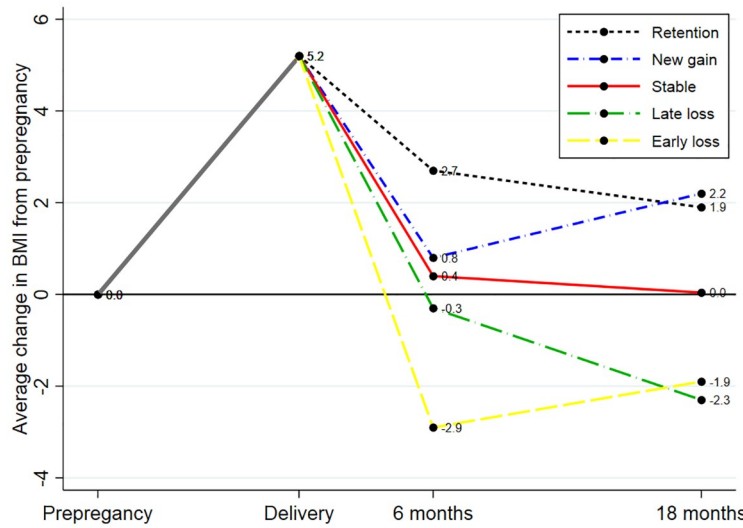

**Fig 2. An illustration of 5 different patterns of postpartum weight change, presented as average change in BMI units in all women from prepregnancy to delivery, 6 and 18 months postpartum.**

Total GWG was obtained from the 6 months postpartum interview and categorized as "below," "within," and "above" the 2009 Institute of Medicine (IOM) recommendations for GWG in their BMI category. It is recommended that women with underweight gain 12.5 to 18 kg, women with normal weight 11.5 to 16 kg, women with overweight 7 to 11.5 kg, and women with obesity 5 to 9 kg (S1 Table) [4].

Furthermore, based on prepregnancy BMI and BMI 6 and 18 months postpartum, we defined 5 different postpartum weight change patterns, which present 2 different ways to obtain an overall weight gain and 2 different ways to obtain an overall weight loss (Fig 2). Women who had an overall weight gain (>1 BMI unit) from before pregnancy to 18 months postpartum were divided into 2 groups: women who had retained weight (6 months weight ≥ 18 months weight) and women who had gained weight postpartum (6 months weight < 18 months weight). Women who had an overall weight loss (<−1 BMI unit) from before pregnancy to 18 months postpartum were divided into 2 groups: women who had an early loss (6 months weight < 18 months weight) and women who had a late loss (6 months weight ≥ 18 months weight). Finally, women who had returned to their prepregnancy BMI at 18 months (±1 BMI unit) were defined as stable in weight.

### Hypertension and CVD

A unique identification number allocated to all residents of Denmark at birth was used for individual linkage to the National Patient Registry. This registry contains information on incident disease diagnoses on all inpatient hospital contacts since 1977, and since 1995, also outpatient and emergency room contacts [23]. We identified incident hospital diagnoses of hypertension (International Classification of Diseases [ICD]-10: I10 and I11), ischemic heart disease (ICD-10: I20, I21, I24, and I25), and stroke (ICD-10: I60 to I64) after 18 months postpartum. Furthermore, we included information on self-reported hypertension from the maternal follow-up. The women reported any hypertension diagnosed by a doctor and the year of the diagnosis. An arbitrary date was set to January 1, as the exact date of diagnosis was unknown. Risk of self-reported hypertension was studied in a subsample of 25,926 women who had no self-reported hypertension before start of follow-up.

## Covariates

We included information from the first pregnancy interview on socio-occupational status (low, middle, or high) [24], alcohol intake before pregnancy (none, 1 to 7, >7 units per week), and overall leisure-time exercise during pregnancy (no exercise, 1 to 180, >180 minutes per week). For smoking status during pregnancy and the first 6 months postpartum (smoking, smoking cessation, or nonsmoking), we also included information from the first postpartum interview. From the food frequency questionnaire, information on dietary patterns was included (Western, intermediate, or health conscious) [25], and the 2 postpartum interviews informed on total breastfeeding duration (<4, 4 to 10, >10 months). The National Patient Registry provided information on diabetes (gestational diabetes mellitus [GDM], pregestational diabetes mellitus, or no diabetes), preeclampsia, and preterm birth occurrence during the index pregnancy. Furthermore, the Danish Medical Birth Registry provided information on parity both before (0, 1, 2+) and after the index pregnancy (0, 1+) [26].

## Statistical methods

All analyses were planned a priori when this study was designed. Cox regression models were used to estimate hazard ratios (HRs) with 95% confidence intervals (CIs) of incident hospital-diagnosed hypertension and CVD according to weight change from prepregnancy to 18 months postpartum. We saw similar associations for ischemic heart disease and stroke, and they were therefore treated as a composite endpoint (CVD) to increase power. Women were considered at risk from the date of second postpartum interview (approximately 18 months postpartum) until the time of diagnosis, emigration, death ($n = 336$ in the hypertension analyses, $n = 348$ in the CVD analyses), or end of follow-up (September 10, 2018), whichever came first. To investigate how GWG may modify the association between overall weight change and risk of hypertension and CVD, we estimated how permutations of GWG (below, within, and above) and overall weight change ($<-1$, $\pm1$, $>1$ BMI units) in 9 categories were associated with both outcomes. Women who gained weight according to the GWG recommendation and had no overall weight change serve as reference. We further examined how the 5 different postpartum weight change patterns were associated with both outcomes. Women who had returned to their prepregnancy BMI at 18 months postpartum served as reference. Finally, we examined whether the associations were modified by prepregnancy BMI ($<25$ and $\geq25$ kg/m$^2$).

The assumption of proportional hazards was examined graphically by log-minus-log plots for the main exposure, and no violation was observed. All analyses were adjusted for a priori selected covariates. We adjusted for BMI, parity, and alcohol intake before the index pregnancy, maternal age at conception, socio-occupational status, dietary intake, leisure-time exercise, diabetes, pre-eclampsia, preterm birth, smoking status during index pregnancy, and total duration of breastfeeding. To be able to evaluate the baseline risk, we estimated adjusted incidence rates using Poisson regression models for a reference woman (characteristics presented in Table 1).

In a sensitivity analysis, we examined risk of incident self-reported hypertension, and findings were similar to those presented (S2 Table). Also, we did a sensitivity analysis adjusted for births (yes/no) during follow-up, and findings were similar to those presented. We further examined our main exposure continuously by restricted cubic splines with 4 knots (fifth, 35th, 65th, and 95th percentiles) and a reference value set to 0 in weight change [27]. The splines supported the findings from the categorical analyses and are presented in S1 Fig.

To address the problem of missing data in covariates, we used multiple imputation [28]. Variables with complete data (prepregnancy weight, height, age at conception, gestational age, and weight 18 months postpartum) were included in the imputation step as explanatory variables in addition to the variables included for imputation. Furthermore, the outcome variable

**Table 1. Maternal characteristics according to weight change from prepregnancy to 18 months postpartum within 47,966 women participating in the DNBC.**

| Variable | Weight change in BMI units from prepregnancy to 18 months postpartum | | | | | | | | Missing |
| | <−1 (n = 9,948) | | −1 to 1 (n = 27,178) | | >1−2 (n = 6,823) | | >2 (n = 4,017) | | |
| | n | % | n | % | n | % | n | % | |
| Age at conception (years) | | | | | | | | | 0 |
| <27 | 2,791 | 28.1 | 6,065 | 22.3 | 1,747 | 25.6 | 1,321 | 32.9 | |
| 27–33 | 4,990 | 50.2 | 14,151 | 52.1 | 3,477 | 51.0 | 1,871 | 46.6 | |
| >33 | 2,167 | 21.8 | 6,962 | 25.6 | 1,599 | 23.4 | 825 | 20.5 | |
| Parity at conception | | | | | | | | | 0 |
| 0 | 4,656 | 46.8 | 11,709 | 43.1 | 3,126 | 45.8 | 2,034 | 50.6 | |
| 1 | 3,675 | 36.9 | 10,509 | 38.7 | 2,553 | 37.4 | 1,311 | 32.6 | |
| 2+ | 1,617 | 16.3 | 4,960 | 18.3 | 1,144 | 16.8 | 672 | 16.7 | |
| Socio-occupational status | | | | | | | | | 153 |
| High | 4,740 | 47.8 | 15,374 | 56.7 | 3,494 | 51.3 | 1,660 | 41.5 | |
| Medium | 4,182 | 42.2 | 9,687 | 35.7 | 2,699 | 39.7 | 1,767 | 44.2 | |
| Low | 987 | 10.0 | 2,038 | 7.5 | 614 | 9.0 | 571 | 14.3 | |
| Prepregnancy BMI (kg/m$^2$) | | | | | | | | | 0 |
| <18.5 | 85 | 0.9 | 1,505 | 5.5 | 361 | 5.3 | 154 | 3.8 | |
| 18.5–24.9 | 4,798 | 48.2 | 20,648 | 76.0 | 4,863 | 71.3 | 2,421 | 60.3 | |
| 25–29.9 | 3,165 | 31.8 | 3,863 | 14.2 | 1,247 | 18.3 | 1,042 | 25.9 | |
| ≥30 | 1,900 | 19.1 | 1,162 | 4.3 | 352 | 5.2 | 400 | 10.0 | |
| Alcohol intake per week before pregnancy (units) | | | | | | | | | 203 |
| 0 | 1,366 | 13.8 | 2,896 | 10.7 | 915 | 13.5 | 715 | 17.9 | |
| >0–7 | 7,633 | 77.1 | 21,480 | 79.3 | 5,262 | 77.5 | 2,962 | 74.2 | |
| >7 | 905 | 9.1 | 2,700 | 10.0 | 614 | 9.0 | 315 | 7.9 | |
| Dietary intake during pregnancy | | | | | | | | | 12,341 |
| Western | 1,345 | 18.3 | 3,286 | 16.2 | 953 | 18.9 | 642 | 21.7 | |
| Intermediate | 4,854 | 66.0 | 13,324 | 65.8 | 3,410 | 67.5 | 1,982 | 67.0 | |
| Health conscious | 1,155 | 15.7 | 3,649 | 18.0 | 689 | 13.6 | 336 | 11.4 | |
| Leisure-time exercise during pregnancy (min/week) | | | | | | | | | 46 |
| None | 6,351 | 63.9 | 16,771 | 61.8 | 4,435 | 65.1 | 2,798 | 69.7 | |
| 1–180 | 2,906 | 29.3 | 8,303 | 30.6 | 1,926 | 28.3 | 938 | 23.4 | |
| >180 | 677 | 6.8 | 2,080 | 7.7 | 456 | 6.7 | 279 | 6.9 | |
| Smoking status during pregnancy and until 6 months postpartum | | | | | | | | | 8,401 |
| Nonsmoking | 5,702 | 68.6 | 16,954 | 76.0 | 4,064 | 72.5 | 2,175 | 65.4 | |
| Smoking cessation | 1,085 | 13.0 | 2,493 | 11.2 | 834 | 14.9 | 664 | 20.0 | |
| Smoking | 1,528 | 18.4 | 2,868 | 12.9 | 711 | 12.7 | 487 | 14.6 | |
| Diabetes during pregnancy | | | | | | | | | 0 |
| None | 9,716 | 97.7 | 26,842 | 98.8 | 6,722 | 98.5 | 3,937 | 98.0 | |
| Pregestational diabetes mellitus | 38 | 0.4 | 72 | 0.3 | 28 | 0.4 | 12 | 0.3 | |
| Gestational diabetes mellitus | 194 | 2.0 | 264 | 1.0 | 73 | 1.1 | 68 | 1.7 | |
| Preeclampsia | | | | | | | | | 0 |
| No | 9,715 | 97.7 | 26,675 | 98.1 | 6,661 | 97.6 | 3,882 | 96.6 | |
| Yes | 233 | 2.3 | 503 | 1.9 | 162 | 2.4 | 135 | 3.4 | |
| Preterm birth | | | | | | | | | 0 |
| No | 9,414 | 94.6 | 25,935 | 95.4 | 6,441 | 94.4 | 3,792 | 94.4 | |
| Yes | 534 | 5.4 | 1,243 | 4.6 | 382 | 5.6 | 225 | 5.6 | |
| GWG according to IOM recommendations | | | | | | | | | 8,689 |
| Below | 1,778 | 21.5 | 4,147 | 18.7 | 670 | 12.0 | 273 | 8.3 | |

(*Continued*)

**Table 1.** (Continued)

| | Weight change in BMI units from prepregnancy to 18 months postpartum | | | | | | | | Missing |
|---|---|---|---|---|---|---|---|---|---|
| | <−1 (n = 9,948) | | −1 to 1 (n = 27,178) | | >1–2 (n = 6,823) | | >2 (n = 4,017) | | |
| Within | 2,804 | 33.9 | 9,171 | 41.4 | 1,911 | 34.3 | 812 | 24.8 | |
| Above | 3,691 | 44.6 | 8,840 | 39.9 | 2,987 | 53.6 | 2,193 | 66.9 | |
| Weight change prepregnancy to 6 months postpartum (BMI units) | | | | | | | | | 9,510 |
| <−1 | 2,186 | 42.4 | 4,443 | 8.7 | 1,370 | 2.9 | 1,056 | 2.2 | |
| −1 to 1 | 3,833 | 43.1 | 11,133 | 64.2 | 2,719 | 33.5 | 1,466 | 16.5 | |
| >1 | 2,435 | 14.5 | 8,033 | 27.1 | 1,822 | 63.6 | 882 | 81.3 | |
| Total breastfeeding duration (months) | | | | | | | | | 6,588 |
| <4 | 3,399 | 25.9 | 1,907 | 18.8 | 156 | 23.2 | 69 | 31.0 | |
| 4–10 | 3,456 | 45.3 | 14,018 | 47.2 | 1,822 | 46.0 | 522 | 43.1 | |
| >10 | 1,165 | 28.8 | 5,913 | 34.0 | 3,457 | 30.8 | 2,572 | 25.9 | |

DNBC, Danish National Birth Cohort; GWG, gestational weight gain; IOM, Institute of Medicine.

for hypertension, ischemic heart disease, and stroke were included together with the Nelson–Aalen estimator, an approximation of the cumulative baseline hazard, as suggested by others [29]. For women still breastfeeding at the time of the interview, total breastfeeding duration was imputed using interval imputation with a lower limit set to the time of the interview and a universal upper limit set to 3 years. A total of 50 copies of the dataset were generated by chained equations. The imputation and subsequent analyses were conducted using standard *mi* procedures available in STATA/SE 15 (StataCorp, College Station, Texas, US). We also carried out complete case analyses and observed results of same direction and approximate magnitude as those presented (S3–S5 Tables).

Finally, death may be a potential competing risk in the present study. Therefore, as suggested in the peer review, we did a sensitivity competing risk analysis of our main exposure using the Fine–Gray approach [30] with death as a competing risk (S6 Table). Results were similar to those observed for the complete case analysis using the Cox regression model.

## Results

During the 16 years of follow-up (median: 16.4; fifth, 95th percentile: 11.3; 18.5), a total of 2,011 women were diagnosed at the hospital with hypertension, 813 with ischemic heart disease and 508 with stroke. At start of follow-up, women were on average 32.3 years old (range 18.0–49.2), and compared with their prepregnancy BMI, 56.7% had a change in BMI within ±1 BMI unit, 20.7% lost >1 BMI unit, 14.2% had gained >1 to 2 BMI unit, and 8.4% had gained >2 BMI units. Women with a stable BMI were more likely to be older, parous, of high socio-occupational status, normal weight, and have had a moderate alcohol intake before the index pregnancy than women who changed BMI. They were also more likely to have had a healthy dietary intake, do exercise, be nonsmokers, and have no gestational diabetes mellitus (GDM), preeclampsia, or preterm birth during the index pregnancy. They breastfed for a longer period and were more likely to have had a GWG within the IOM recommendation and to have returned to their prepregnancy weight by 6 months postpartum (Table 1).

### Weight change from prepregnancy to 18 months postpartum

Weight gains of >1 to 2 BMI units and >2 BMI units from prepregnancy to 18 months postpartum were associated with 25% (95% CI: 10% to 42%), $P = 0.001$ and 31% (14% to 52%),

**Table 2. Adjusted HRs[a] and rates[b] (95% CI) of hypertension and CVD according to weight change from prepregnancy to 18 months postpartum (n = 47,966).**

| | Hypertension | | | | | | CVD | | | | | |
|---|---|---|---|---|---|---|---|---|---|---|---|---|
| | Cases (n) | Rate | 95% CI | HR | 95% CI | P value | Cases (n) | Rate | 95% CI | HR | 95% CI | P value |
| **All** | | | | | | | | | | | | |
| <−1 | 498 | 16.6 | (14.3, 19.3) | 0.98 | (0.88, 1.10) | 0.79 | 325 | 12.4 | (10.3, 15.0) | 1.14 | (0.99, 1.32) | 0.06 |
| −1 to 1 | 964 | 16.8 | (14.8, 19.2) | | Ref | | 656 | 10.9 | (9.2, 12.8) | | Ref | |
| >1 to 2 | 319 | 21.0 | (17.9, 24.6) | 1.25 | (1.10, 1.42) | 0.001 | 174 | 11.1 | (9.0, 13.7) | 1.02 | (0.87, 1.21) | 0.79 |
| >2 | 230 | 22.1 | (18.5, 26.3) | 1.31 | (1.14, 1.52) | <0.001 | 134 | 13.4 | (10.7, 16.8) | 1.24 | (1.02, 1.49) | 0.03 |
| **Prepregnancy BMI <25 kg/m²** | | | | | | | | | | | | |
| <−1 | 141 | 12.3 | (9.7, 15.5) | 0.99 | (0.83, 1.19) | 0.95 | 138 | 11.1 | (8.6, 14.4) | 1.28[c] | (1.06, 1.55) | 0.01 |
| −1 to 1 | 610 | 12.6 | (10.6, 15.1) | | Ref | | 464 | 8.8 | (7.2, 10.8) | | Ref | |
| >1 to 2 | 189 | 15.8 | (12.8, 19.5) | 1.26 | (1.07, 1.48) | 0.006 | 122 | 9.6 | (7.4, 12.4) | 1.09 | (0.89, 1.33) | 0.39 |
| >2 | 112 | 18.5 | (14.4, 23.6) | 1.49 | (1.22, 1.82) | <0.001 | 85 | 12.9 | (9.7, 17.2) | 1.48[d] | (1.17, 1.87) | 0.001 |
| **Prepregnancy BMI ≥25 kg/m²** | | | | | | | | | | | | |
| <−1 | 357 | 30.4 | (34.9, 37.2) | 0.93 | (0.80, 1.08) | 0.37 | 187 | 16.9 | (12.8, 22.4) | 0.95[c] | (0.77, 1.16) | 0.60 |
| −1 to 1 | 354 | 32.4 | (26.5, 39.6) | | Ref | | 192 | 17.8 | (13.5, 23.6) | | Ref | |
| >1 to 2 | 130 | 39.0 | (30.6, 49.7) | 1.21 | (0.99, 1.47) | 0.07 | 52 | 15.4 | (10.7, 22.1) | 0.86 | (0.63, 1.17) | 0.34 |
| >2 | 118 | 36.6 | (28.5, 47.0) | 1.13 | (0.92, 1.39) | 0.25 | 49 | 15.8 | (10.9, 22.9) | 0.88[d] | (0.65, 1.21) | 0.44 |

CI, confidence interval; CVD, cardiovascular disease (ischemic heart disease and stroke); HR, hazard ratio.

a Cox regression models were used to estimate HRs and 95% CIs adjusted for prepregnancy BMI, parity, and alcohol intake before the index pregnancy, maternal age at conception, socio-occupational status, dietary intake, leisure-time exercise, diabetes, preeclampsia, and preterm birth during index pregnancy, smoking status during index pregnancy and the first 6 months postpartum, and total duration of breastfeeding.

b Poisson regression models were used to estimate rates and 95% CIs per 10,000 person-years for a reference woman: primiparous, 29.8 years of age at conception, prepregnancy BMI of 23.5 kg/m² (for BMI <25 kg/m², this was 21.5 kg/m² and for BMI ≥25 kg/m², this was 29.0 kg/m²), high in socio-occupational status, no preeclampsia, no diabetes, delivered at term, and during pregnancy was nonsmoker, had an intermediate dietary pattern, did no exercise, and breastfed total 4 to 10 months.

c P value for interaction by prepregnancy BMI=0.03. The p value refers to the difference between the two associations.

d P value for interaction by prepregnancy BMI=0.01. The p value refers to the difference between the two associations.

$P < 0.001$ higher risks of hypertension compared with a stable BMI (±1 BMI unit) in all women (Table 2). For CVD, an interaction with prepregnancy BMI was observed for the association of weight gain and CVD ($P = 0.01$). Thus, weight gain >2 BMI was associated with 48% (17% to 87%), $P = 0.001$ higher risk of CVD compared with a stable BMI (±1 BMI unit) in women who were normal-/underweight before pregnancy, whereas among women who were overweight/obese before pregnancy, no association was observed (HR 0.88, 95% CI: 0.65; 1.21, $P = 0.44$) (Table 2).

Also, for the association between weight loss and risk of CVD, an interaction with prepregnancy BMI was observed ($P = 0.03$). Thus, compared with women who had a stable BMI (±1 BMI unit), losing >1 BMI unit was associated with 28% (6% to 55%), $P = 0.01$ higher risk of CVD in women with normal-/underweight, whereas in women with overweight/obesity, losing >1 BMI unit was not associated with risk of CVD (HR 0.95, 95% CI: 0.77; 1.16, $P = 0.60$) (Table 2).

## GWG patterns

We observed that within strata of overall weight change, the risk of hypertension was the same, whether the women had gained below, within, or above the GWG recommendations (Table 3). In women with normal-/underweight, the overall weight change from prepregnancy to 18 months postpartum seemed more important in relation to risk of hypertension than

**Table 3. Adjusted HRs[a] (95% CI) of hypertension and CVD according to adherence to the IOM recommendations for GWG and weight change from prepregnancy to 18 months postpartum, $n$ = 47,966.**

| | Hypertension | | | | | | | | | CVD | | | | | | | | |
| | GWG recommendations | | | | | | | | | GWG recommendations | | | | | | | | |
| | Below | | | Within | | | Above | | | Below | | | Within | | | Above | | |
| Weight change prepregnancy to 18 months postpartum (BMI units) | HR | 95% CI | P value | HR | 95% CI | P value | HR | 95% CI | P value | HR | 95% CI | P value | HR | 95% CI | P value | HR | 95% CI | P value |
|---|---|---|---|---|---|---|---|---|---|---|---|---|---|---|---|---|---|---|
| **All** | $n = 8,578$ | | | $n = 17,188$ | | | $n = 22,200$ | | | $n = 8,578$ | | | $n = 17,188$ | | | $n = 22,200$ | | |
| <−1 | 1.04 | (0.83, 1.30) | 0.73 | 1.00 | (0.82, 1.23) | 0.999 | 0.93 | (0.77, 1.12) | 0.42 | 1.38 | (1.07, 1.80) | 0.02 | 1.03 | (0.79, 1.33) | 0.85 | 1.15 | (0.91, 1.44) | 0.24 |
| −1 to 1 | 1.00 | (0.82, 1.22) | 0.98 | | Ref | | 1.00 | (0.86, 1.17) | 0.98 | 0.92 | (0.73, 1.17) | 0.51 | | Ref | | 1.08 | (0.89, 1.30) | 0.42 |
| >1 | 1.27 | (0.94, 1.71) | 0.11 | 1.37 | (1.12, 1.67) | 0.002 | 1.23 | (1.05, 1.45) | 0.01 | 0.97 | (0.64, 1.48) | 0.90 | 0.97 | (0.74, 1.26) | 0.80 | 1.24 | (1.02, 1.52) | 0.03 |
| **Prepregnancy BMI <25 kg/m²** | $n = 7,135$ | | | $n = 14,154$ | | | $n = 13,546$ | | | $n = 7,135$ | | | $n = 14,154$ | | | $n = 13,546$ | | |
| <−1 | 1.10 | (0.80, 1.52) | 0.54 | 0.81 | (0.58, 1.14) | 0.23 | 1.07 | (0.76, 1.48) | 0.71 | 1.70[b] | (1.24, 2.35) | 0.001 | 0.96 | (0.66, 1.38) | 0.82 | 1.23 | (0.86, 1.76) | 0.25 |
| −1 to 1 | 1.09 | (0.87, 1.36) | 0.47 | | Ref | | 0.99 | (0.80, 1.21) | 0.89 | 0.99 | (0.76, 1.29) | 0.96 | | Ref | | 1.03 | (0.81, 1.30) | 0.83 |
| >1 | 1.44 | (1.02, 2.02) | 0.036 | 1.40 | (1.10, 1.78) | 0.007 | 1.28 | (1.02, 1.59) | 0.03 | 1.04 | (0.66, 1.65) | 0.86 | 1.09 | (0.81, 1.47) | 0.57 | 1.38[c] | (1.08, 1.76) | 0.01 |
| **Prepregnancy BMI ≥25 kg/m²** | $n = 1,443$ | | | $n = 3,034$ | | | $n = 8,654$ | | | $n = 1,443$ | | | $n = 3,034$ | | | $n = 8,654$ | | |
| <−1 | 0.91 | (0.66, 1.26) | 0.56 | 0.94 | (0.70, 1.25) | 0.67 | 0.74 | (0.56, 0.97) | 0.027 | 0.92[b] | (0.58, 1.45) | 0.71 | 0.88 | (0.58, 1.32) | 0.53 | 0.88 | (0.62, 1.26) | 0.48 |
| −1 to 1 | 0.80 | (0.52, 1.23) | 0.31 | | Ref | | 0.86 | (0.66, 1.11) | 0.25 | 0.71 | (0.39, 1.31) | 0.28 | | Ref | | 0.97 | (0.68, 1.37) | 0.85 |
| >1 | 0.89 | (0.48, 1.64) | 0.71 | 1.23 | (0.86, 1.76) | 0.26 | 1.00 | (0.77, 1.30) | 0.99 | 0.76 | (0.30, 1.92) | 0.56 | 0.58 | (0.31, 1.09) | 0.09 | 0.89[c] | (0.62, 1.28) | 0.52 |

CI, confidence interval; CVD, cardiovascular disease (ischemic heart disease and stroke); GWG, gestational weight gain; HR, hazard ratio; IOM, Institute of Medicine.

a Cox regression models were used to estimate HRs and 95% CIs adjusted for prepregnancy BMI, parity, and alcohol intake before the index pregnancy, maternal age at conception, socio-occupational status, dietary intake, leisure-time exercise, diabetes, preeclampsia, and preterm birth during index pregnancy, smoking status during index pregnancy and the first 6 months postpartum, and total duration of breastfeeding.

b P value for interaction by prepregnancy BMI=0.03. The p value refers to the difference between the two associations.

c P value for interaction by prepregnancy BMI=0.048. The p value refers to the difference between the two associations.

GWG. Compared with women with normal-/underweight who had returned to their prepregnancy BMI by 18 months postpartum and gained within the GWG recommendation, women with normal-/underweight who gained >1 BMI unit from prepregnancy to 18 months postpartum and gained above the GWG recommendation had a 28% (2% to 59%), $P$ = 0.03 higher risk of hypertension, which was 40% (10% to 78%), $P$ = 0.007 with GWG within recommended, and 44% (2% to 102%), $P$ = 0.036 with GWG below recommended. No such associations were observed in women who were overweight/obese, but test for interaction did not reach statistical significance (Table 3).

For CVD, risks differed by adherence to GWG recommendations and prepregnancy BMI (Table 3). Compared with women with normal/underweight who had returned to their prepregnancy BMI at 18 months postpartum and gained within the GWG recommendation, women with normal-/underweight who gained >1 BMI unit from prepregnancy to 18 months postpartum and above the GWG recommendation had 38% (8% to 76%), $P$ = 0.01 increased risk of CVD; no such association was observed in women with overweight/obesity ($P$ interaction = 0.048). Women who were normal-/underweight and lost >1 BMI unit from prepregnancy to 18 months postpartum and gained below the GWG recommendation had 70% (24% to 135%), $P$ = 0.001 increased risk of CVD; this was also not observed in women with overweight/obesity ($P$ interaction = 0.03).

## Postpartum weight change patterns

The 5 groups of different postpartum weight change patterns including their average BMI change from prepregnancy to 6 and 18 months postpartum are presented in Fig 2.

Compared with women who had a stable weight postpartum, women who gained >1 BMI unit from prepregnancy to 18 months postpartum by a weight retention postpartum had 28% (11% to 48%), *P* = 0.001 increased risk of hypertension and women who had a new gain postpartum had 26% (10% to 45%), *P* = 0.001 increased risk of hypertension. These excess risks were slightly higher for women with normal-/underweight than for women with overweight/obesity (Table 4). For CVD, we observed modest excess risks related to both postpartum weight retention or a new gain postpartum in women with normal-/underweight. This was not seen in women with overweight/obesity.

For women who lost >1 BMI unit from prepregnancy to 18 months postpartum, losing substantial weight early, i.e., from prepregnancy to 6 months postpartum, increased risk of hypertension by 27% (2% to 59%), *P* = 0.04 and CVD by 42% (7% to 89%), *P* = 0.02 in all women compared with women who had a stable weight postpartum. In contrast, a late weight loss, i.e., from 6 to 18 months postpartum, only increased risk of CVD among women with normal-/underweight (HR 1.27, 95% CI: 1.03; 1.56, *P* = 0.02) and not among women with overweight/obesity (HR 0.88, 95% CI: 0.70; 1.09, *P* = 0.24) (Table 4).

## Discussion

In this large cohort study with 16 years of follow-up, we found that weight gain of 1 BMI unit or more from before pregnancy to 18 months postpartum was associated with a higher risk of hypertension in all women and a higher risk of CVD in women with normal-/underweight. Whether the gain was caused by postpartum weight retention or a new weight gain postpartum did not matter. Neither did risk of hypertension depend on whether the woman had gained below, within, or above the GWG recommendations. Weight loss from before pregnancy to 18 months postpartum was associated with increased risk of CVD in women with normal-/underweight, especially when they had GWG below the recommendation. Such increased risk observed with a weight loss was not seen among women with overweight/obesity.

Higher weight postpartum than before pregnancy or early in pregnancy is associated with greater risk of obesity, more abdominal adiposity [6,8,12,31,32], and a more atherogenic lipid profile [33] which may explain our observed increased risk of hypertension and CVD with a weight gain. Further, others have observed that women who gained weight from 3 to 12 months postpartum have higher blood pressure, greater insulin resistance, lower adiponectin, and higher low-density lipoprotein (LDL) cholesterol than women who lost weight in the same period [34]. This support our findings of an increased risk of hypertension with a weight gain from 6 to 18 months postpartum compared with a stable weight. The more distinct associations we observed in women with normal-/underweight than in women with overweight/obese may be due to a higher underlying baseline risk observed in the latter group as also observed by others [19]. Also, our overweight/obese group covered a wide range of BMI and baseline risks of CVD that may limit our ability to study weight change in this group. Very few studies have been done on maternal weight change related to childbearing and later cardiovascular health. One small study showed an increased risk of heart disease, hypertension, and dyslipidemia with long-term weight gain from early gestation to 15 years after delivery [35]. In agreement with our findings, others have observed that GWG above recommended levels was not associated with elevated blood pressure 4 to 7 years after delivery [36]. However, they did not consider postpartum weight change, and we observed that postpartum weight may be more important than GWG. On the other hand, another study showed that GWG in the first trimester was positively associated with blood pressure 7 years after delivery, but they also concluded that first trimester GWG was most strongly associated with greater weight postpartum [37]. One must be aware that GWG recommendations are prepregnancy BMI specific, and

**Table 4. Adjusted HRs[a] (95% CI) of hypertension and CVD according to weight change patterns from prepregnancy to 18 months postpartum (n = 47,966).**

| Weight change prepregnancy to 18 months postpartum (BMI units) | Postpartum weight change pattern[b] | n | Hypertension HR | Hypertension 95% CI | Hypertension P value | CVD HR | CVD 95% CI | CVD P value |
|---|---|---|---|---|---|---|---|---|
| **All** | | | | | | | | |
| <−1 | Early loss | 1,488 | 1.27 | (1.02, 1.59) | 0.04 | 1.42 | (1.07, 1.89) | 0.02 |
|  | Late loss | 8,460 | 0.93 | (0.82, 1.06) | 0.27 | 1.09 | (0.94, 1.27) | 0.25 |
| −1 to 1 | Stable | 27,178 | | Ref | | | Ref | |
| >1 | Retention | 5,214 | 1.28 | (1.11, 1.48) | 0.001 | 1.09 | (0.90, 1.32) | 0.38 |
|  | New gain | 5,626 | 1.26 | (1.10, 1.45) | 0.001 | 1.12 | (0.93, 1.34) | 0.23 |
| **Prepregnancy BMI <25 kg/m$^2$** | | | | | | | | |
| <−1 | Early loss | 647 | 1.47 | (0.97, 2.22) | 0.07 | 1.36 | (0.82, 2.26) | 0.23 |
|  | Late loss | 4,236 | 0.92 | (0.75, 1.13) | 0.43 | 1.27 | (1.03, 1.56) | 0.02 |
| −1 to 1 | Stable | 22,153 | | Ref | | | Ref | |
| >1 | Retention | 3,957 | 1.32 | (1.10, 1.59) | 0.003 | 1.13 | (0.90, 1.42) | 0.29 |
|  | New gain | 3,842 | 1.35 | (1.12, 1.63) | 0.001 | 1.32 | (1.06, 1.64) | 0.02 |
| **Prepregnancy BMI ≥25 kg/m$^2$** | | | | | | | | |
| <−1 | Early loss | 840 | 1.16 | (0.88, 1.51) | 0.29 | 1.29 | (0.90, 1.84) | 0.16 |
|  | Late loss | 4,225 | 0.89 | (0.76, 1.04) | 0.15 | 0.88 | (0.70, 1.09) | 0.24 |
| −1 to 1 | Stable | 5,025 | | Ref | | | Ref | |
| >1 | Retention | 1,256 | 1.21 | (0.96, 1.52) | 0.10 | 0.98 | (0.70, 1.36) | 0.88 |
|  | New gain | 1,785 | 1.14 | (0.93, 1.39) | 0.20 | 0.80 | (0.58, 1.09) | 0.15 |

CI, confidence interval; CVD, cardiovascular disease (ischemic heart disease and stroke); HR, hazard ratio.

a Cox regression models were used to estimate HRs and 95% CIs adjusted for prepregnancy BMI, parity, and alcohol intake before the index pregnancy, maternal age at conception, socio-occupational status, dietary intake, leisure-time exercise, diabetes, preeclampsia, and preterm birth during index pregnancy, smoking status during index pregnancy and the first 6 months postpartum, and total duration of breastfeeding.

b Indicates how the overall weight change from prepregnancy to 18 months postpartum was reached by including weight 6 months postpartum: Early loss: 6 months weight < 18 months weight; Late loss: 6 months weight ≥ 18 months weight; New gain: 6 months weight < 18 months weight; and Retention: 6 months weight ≥ 18 months weight.

other results may have been observed with equivalent GWG groups for all women independently of their prepregnancy BMI.

Weight loss may not be beneficial to CVD risk. In healthy individuals, weight loss, intended or unintended, has been associated with higher mortality [38,39], and short-term weight loss over 3 years in middle-aged men and women has been related to a higher risk of coronary heart disease and stroke [40]. Likewise, we observed that maternal weight loss from prepregnancy to 18 months postpartum increased risk of CVD in mothers who were normal-/underweight before pregnancy and gained less than recommended during pregnancy. Among all women, we also observed an increased risk of hypertension and CVD with a 6 months postpartum weight that was substantially below prepregnancy weight. Gaining too little during pregnancy and having a substantially lower weight than before pregnancy by 6 months postpartum may result from a complicated pregnancy. Inadequate GWG is associated with a higher risk of having a small-for-gestational age baby and preterm deliveries [41], which on the other hand are shown to be related to later maternal subclinical and clinical CVD [42,43]. The existing evidence suggests that this may be explained by common predisposing factors for both pregnancy complications and CVD. Thus, it is possible that some sort of unknown confounding generates the association or that reverse causality may be in play with subclinical CVD inducing weight loss, followed by

subsequent clinically manifest CVD. We did not observe an increased risk of hypertension or CVD when overall weight loss was characterized by a substantial weight loss from 6 to 18 months postpartum in women with overweight/obesity, which was likely achieved from an active and intentional change in behavior to attain a postpartum weight loss. However, we did so in women with normal-/underweight for risk of CVD. These findings are supported by studies that show beneficial effects on mortality of intended or unintended weight loss in obese populations [44], but not in nonobese and healthy populations [45,46]. Low fat-free mass [47,48] and, in elderly people [49], loss of fat-free mass is associated with increased mortality. It is likely that women with normal-/underweight who lost weight from before pregnancy to 18 months postpartum had lost not only fat mass but also fat-free mass, with an adverse effect on their cardiovascular health.

## Strengths and limitations

The strengths of our study include our ability to study maternal weight changes by using prospectively collected maternal report of weights which reduces risk of recall bias. Further, the repeated weight measures, including both 6 and 18 months postpartum, made it possible to study different weight patterns, which is important as women may experience very different weight patterns in pregnancy and early motherhood. Also, the large cohort and the extended follow-up period enabled us to study risk of hypertension and CVD among premenopausal women, thus before the hormonal changes of menopause may cause a greater occurrence of these diseases. Our study is further strengthened by the use of national register–based data on the occurrence of hypertension and CVD, which ensures full follow-up and thereby limits the risk of selection bias. Some of the cardiovascular diagnoses in the National Patient Registry have been validated and, for women, positive predictive values of 98% for hypertension and 97% for myocardial infarction have been observed [50]. The equivalent percentage for stroke was 79% [51]. However, we do not believe that any misclassification of our outcome may relate to weight change, thus any bias is likely to cause a potential attenuation of the associations. Hospital-diagnosed hypertension may underestimate the total incidence of hypertension as hypertension is often diagnosed by the general practitioner; however, we were able to analyze self-reported incident hypertension during the follow-up period. We had 50% more self-reported hypertension diagnoses than hospital-diagnosed hypertension, and we observed similar associations with the 2 outcomes.

Death is a competing risk when we study the occurrence of hypertension and CVD. However, the analyses estimating HRs by the Cox regression approach remain valid irrespective of the competing risk, although it is important to note that a higher hazard for cardiovascular events in 1 group may not translate into a higher probability (cumulative incidence) of observing the event in that group [52]. Our sensitivity competing risk analysis did, however, show that HRs based on sub-distribution hazards (Fine–Gray approach [30]) gave virtually identical results. Thus, despite the presence of competing risks, increased hazards can be interpreted as leading also to an increased risk of hypertension and CVD within the age range observed in this study. This is also a consequence of the relative low number of deaths (0.7%) observed within the follow-up period.

Further, a potential limitation of our study is the use of self-reported weights, which may cause some misclassification as women often underreport their weight, and this occurs among obese women to a greater extent [53]. However, as we examined differences in weight within the same person, we believe that this may reduce such misclassification and that it is unlikely to be related to later hypertension and CVD occurrence. Also, we are unaware of the mother's weight patterns during the follow-up period which may affect the associations. Furthermore,

although we were able to adjust for several potential confounders, we cannot rule out the presence of residual or unmeasured confounding. The mother's weight patterns before pregnancy, body composition, and metabolic health may be potential confounders, as they may affect both her weight change during pregnancy and after birth as well as her risk of hypertension and CVD. We restricted our study population to women who had no hypertension or CVD diagnosis prior to start of follow-up, but metabolic disorders may still be present. Future studies are needed that include information on maternal prepregnancy metabolic status and weight history which will help us to elucidate the specific influence of postpartum weight change on later cardiovascular health. However, our findings suggest that mother's postpartum weight changes also influence her long-term risk of hypertension and CVDs. Thus, our findings support the concept that healthcare providers should expand their existing focus on pregnant women's weight to include postpartum weight.

## Conclusions

In conclusion, we showed that women's cardiovascular health in early and middle adulthood may be affected by their previous weight change from before pregnancy to 18 months postpartum. Thus, women who gained weight throughout this period may have an increased risk of hypertension, and among women with normal-/underweight, also an increased risk of CVD. At the same time, losing weight was associated with increased risk of CVD in women with normal-/underweight, especially among those who gained below the GWG recommendations; this association was not observed among women with overweight/obesity. Our findings emphasize the importance of focusing on mothers' weight patterns postpartum to improve long-term cardiovascular health.

## Supporting information

**S1 STROBE Checklist. STROBE, Strengthening the Reporting of Observational Studies in Epidemiology.**
(DOC)

**S1 Table. The 2009 IOM's recommendations for GWG according to prepregnancy BMI category.** GWG, gestational weight gain; IOM, Institute of Medicine.
(DOCX)

**S2 Table. Adjusted[a] HRs and rates (95% CI) of self-reported hypertension according to weight change from prepregnancy to 18 months postpartum (*n* = 25,926).** CI, confidence interval; HR, hazard ratio.
(DOCX)

**S3 Table. Adjusted HRs[a] and rates[b] (95% CI) of hypertension and CVD according to weight change from prepregnancy to 18 months postpartum (*n* = 27,645)—complete case analyses.** CI, confidence interval; CVD, cardiovascular disease; HR, hazard ratio.
(DOCX)

**S4 Table. Adjusted HRs[a] (95% CI) of hypertension and CVD according to adherence to the IOM recommendations for GWG and weight change from prepregnancy to 18 months postpartum (*n* = 27,449)—complete case analyses.** CI, confidence interval; CVD, cardiovascular disease; GWG, gestational weight gain; HR, hazard ratio; IOM, Institute of Medicine.
(DOCX)

**S5 Table. Adjusted HRs[a] (95% CI) of hypertension and CVD according to weight change patterns from prepregnancy to18 months postpartum (*n* = 27,230)—complete case**

**analyses.** CI, confidence interval; CVD, cardiovascular disease; HR, hazard ratio.
(DOCX)

**S6 Table. Adjusted sub-distribution HRs[a] (95% CI) of hypertension and CVD according to weight change from prepregnancy to 18 months postpartum ($n$ = 27,645)—complete case analyses.** CI, confidence interval; CVD, cardiovascular disease; HR, hazard ratio.
(DOCX)

**S1 Fig.** Adjusted HRs (95% CI) of hypertension (A) and CVD (B) in relation to weight change from prepregnancy to 18 months postpartum in BMI units (all women; women with a prepregnancy BMI <25; and women with a prepregnancy BMI ≥25). Adjusted for maternal age at conception, socio-occupational status, parity, prepregnancy BMI, alcohol intake before the index pregnancy and dietary intake, leisure-time exercise, diabetes, preeclampsia, and preterm birth during index pregnancy, smoking status during index pregnancy and the first 6 months postpartum, and total duration of breastfeeding. CI, confidence interval; CVD, cardiovascular disease; HR, hazard ratio.
(DOCX)

## Author Contributions

**Conceptualization:** Helene Kirkegaard, Kathleen M. Rasmussen, Erica P. Gunderson, Lars Køber, Thorkild I. A. Sørensen, Ellen A. Nøhr.

**Data curation:** Helene Kirkegaard, Mette Bliddal.

**Formal analysis:** Helene Kirkegaard, Mette Bliddal, Henrik Støvring, Ellen A. Nøhr.

**Funding acquisition:** Helene Kirkegaard.

**Investigation:** Helene Kirkegaard, Mette Bliddal, Henrik Støvring, Kathleen M. Rasmussen, Erica P. Gunderson, Lars Køber, Thorkild I. A. Sørensen, Ellen A. Nøhr.

**Methodology:** Helene Kirkegaard, Mette Bliddal, Henrik Støvring, Kathleen M. Rasmussen, Erica P. Gunderson, Lars Køber, Thorkild I. A. Sørensen, Ellen A. Nøhr.

**Project administration:** Helene Kirkegaard.

**Supervision:** Henrik Støvring, Thorkild I. A. Sørensen, Ellen A. Nøhr.

**Writing – original draft:** Helene Kirkegaard.

**Writing – review & editing:** Helene Kirkegaard, Mette Bliddal, Henrik Støvring, Kathleen M. Rasmussen, Erica P. Gunderson, Lars Køber, Thorkild I. A. Sørensen, Ellen A. Nøhr.

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
