## [Editor Report · Decision Letter 0]

30 Jan 2020

Dear Dr Kirkegaard, 

Thank you for submitting your manuscript entitled "Maternal weight change from prepregnancy to 18 months postpartum and subsequent risk of hypertension and cardiovascular disease: A study within the Danish National Birth Cohort" for consideration by PLOS Medicine.

Your manuscript has now been evaluated by the PLOS Medicine editorial staff [as well as by an academic editor with relevant expertise] and I am writing to let you know that we would like to send your submission out for external peer review.

Kind regards,

Adya Misra, PhD,

Senior Editor

PLOS Medicine

---

## [Decision Letter · Decision Letter 1]

12 May 2020

Dear Dr. Kirkegaard,

Thank you very much for submitting your manuscript "Maternal weight change from prepregnancy to 18 months postpartum and subsequent risk of hypertension and cardiovascular disease: A study within the Danish National Birth Cohort" (PMEDICINE-D-19-04657R1) for consideration at PLOS Medicine. 

[LINK]

In light of these reviews, I am afraid that we will not be able to accept the manuscript for publication in the journal in its current form, but we would like to consider a revised version that addresses the reviewers' and editors' comments. Obviously we cannot make any decision about publication until we have seen the revised manuscript and your response, and we plan to seek re-review by one or more of the reviewers. 

We expect to receive your revised manuscript by May 26 2020 11:59PM. Please email us (plosmedicine@plos.org) if you have any questions or concerns.

We look forward to receiving your revised manuscript. 

Sincerely,

Adya Misra, PhD

Senior Editor 

PLOS Medicine

plosmedicine.org

Abstract 

* Please structure your abstract using the PLOS Medicine headings (Background, Methods and Findings, Conclusions).

* Please combine the Methods and Findings sections into one section, “Methods and findings”.

* Please include the study design, population and setting, number of participants, years during which the study took place, length of follow up, and main outcome measures. 

* Please quantify the main results (with 95% CIs and p values).

References- in square brackets and bibliography formatting in Vancouver style

Please provide citations to, or copies of questionnaires and interview guides used in the study 

Did your study have a prospective protocol or analysis plan? Please state this (either way) early in the Methods section. a) If a prospective analysis plan (from your funding proposal, IRB or other ethics committee submission, study protocol, or other planning document written before analyzing the data) was used in designing the study, please include the relevant prospectively written document with your revised manuscript as a Supporting Information file to be published alongside your study, and cite it in the Methods section. A legend for this file should be included at the end of your manuscript. b) If no such document exists, please make sure that the Methods section transparently describes when analyses were planned, and when/why any data-driven changes to analyses took place. c) In either case, changes in the analysis-- including those made in response to peer review comments-- should be identified as such in the Methods section of the paper, with rationale.

Please ensure that the study is reported according to the [STROBE] guideline, and include the completed [STROBE or other] checklist as Supporting Information. When completing the checklist, please use section and paragraph numbers, rather than page numbers. Please add the following statement, or similar, to the Methods: "This study is reported as per the Strengthening the Reporting of Observational Studies in Epidemiology (STROBE) guideline (S1 Checklist)."

On Page 9, second paragraph you have included an instance of “data not shown”, which is not permitted as per PLOS data policy. Please remove this instance or include the data as main text or SI files. 

On page 10, please introduce the term GDM on first view 

Please provide 95% CI and p values whenever reporting numerical results or as needed

Page 11 last paragraph states “GWG recommendation had 70% (24-235%)”. Please correct and clarify as needed

Please present and organize the Discussion as follows: a short, clear summary of the article's findings; what the study adds to existing research and where and why the results may differ from previous research; strengths and limitations of the study; implications and next steps for research, clinical practice, and/or public policy; one-paragraph conclusion.

Comments from the reviewers:

Reviewer #1: Comments to the authors 

This study investigates maternal weight change from pre-pregnancy to 18 months postpartum and subsequent risk of hypertension and cardiovascular disease. 

This study is well-written, clearly described, include relevant tables and figures, and are of highly clinical relevance. Hence, I have very few comments. Although the self-reported weights are a limitation (as acknowledged by the authors) the fact that the authors have access to 18 months postpartum weights are a great strength that balance up the limitations of the self-reported weights. 

As I understand from the submission system this article has already undergone one round of revision (?) but as I was not involved in the first round nor have seen the other reviewers' comments, my comments are totally new and not related to any previous comments. 

1. Missing data: In the method section the authors describe that they are using multiple imputation for imputation of covariates with missing values. However, they are not imputed the values for those with missing pre-pregnancy BMI as these instead are excluded from the cohort, why are not the values of the 5743 missing BMI (i.e. corresponding to 8% missing) imputed?

2. Association of losing weight and increased risk of CVD among women with a BMI <25

The study concludes that the women with a BMI<25 that have a weight loss of 1 unit in BMI from pre-pregnancy until 18 mo postpartum may have an increased risk of CVD, and this association foremost seen in the group of women that also gain below the IOM guidelines in weight during pregnancy. 

The mechanisms are somewhat discussed in the discussion, but I wonder if the authors have further looked into this association and tried to find an explanation? Are the numbers sufficient to stratify BMI further, i.e. separate underweight and normal weight - could it be that this relationship could be modified by BMI further, i.e. is there a stronger relationship among underweight for example. There could also be unmeasured confounding, which the authors also discussed somewhat. Further, as the follow-up time is 16 years there could also be other factors /weight loss/wegiht gain during this period of time that could play an important role. Perhaps this could be discussed even further in the discussion. 

Minor 

Abstract - New weight gain is a bit hard to understand when reading only the abstract, hence adding the time-period to define new gainers would be beneficial

Figure 2 - This figure is beautiful and clearly describes the different weight patterns, however it would benefit much by adding different colors to the lines. 

Reviewer #2: This manuscript examines weight changes among the Danish National Birth Cohort from pre-pregnancy to 18 months postpartum and the associated with the incidence of cardiovascular disease and hypertension. Cardiovascular disease is one of the leading causes of death among women around the world and pregnancy and the postpartum period represent a critical time for identifying women who may be at increased risk and for implementing risk reduction strategies. A challenge for clinicians and researchers working in this field is the male-centric nature of many of our available risk identification tools. In this manuscript the authors identify and explore female-specific risk factors for hypertension and cardiovascular disease. This is valuable work that adds significant information to this field of research. 

The methods used in the analysis are robust, appropriate and well described.

With regards to outcomes, the secondary analysis including self-identified hypertension diagnosis strengthens the results, considering that hypertension is often a primary care diagnosis. 

While examining only codes related to hypertension, ischemic heart disease and stroke is valid, I wonder whether a broader definition of cardiovascular disease, such as the composite outcome included in the CANHEART Study, would strengthen the results of this analysis. Could the authors elaborate on their reasoning for including their specific set of outcomes?

[Tu JV, C.A., Donovan LR, Ko DT, Booth GL, Tu K, et al., The Cardiovascular Health in Ambulatory Care Research Team (CANHEART): Using Big Data to Measure and Improve Cardiovascular Health and Healthcare Services. Circulation: Cardiovascular Quality and Outcomes, 2015. 8(2): p. 204-212.]

With regards to covariates, could the authors specify whether the intensity of exercise was considered? 

When controlling for pregnancy complications, was gestational hypertension considered? Or only preeclampsia?

Did the authors consider adjusting for exposure to pregnancy complications in pregnancies prior to the index pregnancy?

Were other placental disorders, such as abruption, considered? 

The authors have appropriately addressed the limitations of this work and highlighted the strengths. 

The conclusions appear to be supported by the results presented. 

This study addresses an important issue and provides information that may help to improve women's long term health and primary preventative care. 

Reviewer #3: This is a well-conducted study on the association between maternal weight change from prepregnancy to 18 months postpartum and subsequent risk of hypertension and cardiovascular disease using a large national birth cohort. The study design, datasets, statistical methods and analyses, and presentation (tables and figures) and interpretation of results are mostly adequate. However, there are still a few issues needing attention.

1) In statistical methods, the authors said "to address the problem of missing data in covariates, we used multiple imputation". It seems that all the analyses with missing data were automatically done with multiple imputation. However, there are always some bias when imputing data. Researcher normally do this in two steps, using the complete data for the formal analyses and then use imputed data for sensitivity analyses. Can authors please follow this good practice if possible? Quite a few variables with 20-30% missing data are still in the analyses which is a bit worrying. The other way to get around this is to create a dummy category of 'unknown' for those missing data so all the data can be analysed.

2) Table 1 should appear in supplementary information as just recommendations.

3) Table 2. All the categorical variables should be presented with count (percentage) rather than just percentage.

4) Table 3. All the 95% CI for rates and HRs should be (xx, yy) not (xx; yy). Please replace ";" with ",". Also, in the footnote, please mention the statistical methods used to derive these rates and HRs. The same also applies to Table 4 and 5.

5) How many deaths were recorded during the 16-years follow up? How were they treated in the analyses? This should be discussed as there is potentially a competing risk from death when the outcomes are hypertension and CVD.

[LINK]

---

## [Decision Letter · Decision Letter 2]

20 Jul 2020

Dear Dr. Kirkegaard,

Thank you very much for submitting your manuscript "Maternal weight change from prepregnancy to 18 months postpartum and subsequent risk of hypertension and cardiovascular disease: A study within the Danish National Birth Cohort" (PMEDICINE-D-19-04657R2) for consideration at PLOS Medicine. 

[LINK]

In light of these reviews, I am afraid that we will not be able to accept the manuscript for publication in the journal in its current form, but we would like to consider a revised version that addresses the reviewers' and editors' comments. Obviously we cannot make any decision about publication until we have seen the revised manuscript and your response, and we plan to seek re-review by one or more of the reviewers. Specifically, the comments from the statistical reviewer need to be addressed. 

We expect to receive your revised manuscript by Aug 10 2020 11:59PM. Please email us (plosmedicine@plos.org) if you have any questions or concerns.

We look forward to receiving your revised manuscript. 

Sincerely,

Adya Misra, PhD

Senior Editor 

PLOS Medicine

plosmedicine.org

Comments from the reviewers:

Reviewer #2: The authors have appropriately addressed all issues raised by the reviewers. This study addresses an important issue and provides information that may help to improve women's long term health and primary preventative care. Methods, strengths and limitations are all well outlined. In my opinion this article is suitable for publication. 

Reviewer #3: Thanks authors for their effort to improve the manuscript. However, there are still two remaining issues needing attention.

1) On multiple imputation: it says "we carried out a complete case analysis of...". Can authors please provide the complete case analyses result table in the supplementary information?

2) On competing risk: the numbers of death are not small especially for CVD analyses, there are around 1200 CVD cases while 348 deaths happened. Censoring will not solve the competing risk problem. Therefore, competing risk analyses need to be carried out. Also, the authors do need to discuss the impact of competing risk in the main text rather than just in the response.

[LINK]

---

## [Decision Letter · Decision Letter 3]

10 Nov 2020

Dear Dr. Kirkegaard,

Thank you very much for re-submitting your manuscript "Maternal weight change from prepregnancy to 18 months postpartum and subsequent risk of hypertension and cardiovascular disease: A study within the Danish National Birth Cohort" (PMEDICINE-D-19-04657R3) for review by PLOS Medicine.

I have discussed the paper with my colleagues and the academic editor and it was also seen again by xxx reviewers. I am pleased to say that provided the remaining editorial and production issues are dealt with we are planning to accept the paper for publication in the journal.

[LINK]

We look forward to receiving the revised manuscript by Nov 17 2020 11:59PM. 

Sincerely,

Adya Misra, PhD

Senior Editor 

PLOS Medicine

plosmedicine.org

Requests from Editors:

Title: Suggest revising to “Maternal weight change from prepregnancy to 18 months postpartum and subsequent risk of hypertension and cardiovascular disease in Danish women: A Cohort study ”

The abstract limitations should be explicit, for example starting with “the limitations of this work include ..” or similar. Please provide 2-3 limitations here

Please provide brief participant demographics in the abstract

Please add numbers of CVD events in the abstract, from around line 201

We suggest removing the word “prospective”. We think that you are reporting a retrospective analysis of a prospectively gathered dataset, and ask that you adapt the wording at line 5 - and any other relevant instances in the paper - accordingly

a) If a prespecified analysis plan (from your funding proposal, IRB or other ethics committee submission, study protocol, or other planning document written before analyzing the data) was used in designing the study, please include the relevant prospectively written document with your revised manuscript as a Supporting Information file to be published alongside your study, and cite it in the Methods section. A legend for this file should be included at the end of your manuscript.

Please change all iterations of “overweight” and “obese” to “with overweight” and “with obesity” in line with principles of people first language

Please provide access details for Ref 53?

Please provide a completed STROBE checklist as supplementary information. When completing the checklist, please use section and paragraph numbers, rather than page numbers.

"data are" in the data statement

Comments from Reviewers:

Reviewer #3: Thanks authors for their effort to improve the manuscript. The authors have addressed my comments comprehensively. I am satisfied with the response and the revision. No further issues needing attention.

[LINK]

---

## [Editor Report · Decision Letter 4]

17 Mar 2021

Dear Dr. Kirkegaard,

I am writing concerning your manuscript submitted to PLOS Medicine, entitled “Maternal weight change from prepregnancy to 18 months postpartum and subsequent risk of hypertension and cardiovascular disease in Danish women: A cohort study”.

We have now completed our final technical checks and have approved your submission for publication. You will shortly receive a letter of formal acceptance from the editor.

Kind regards,

PLOS Medicine